# Impact of Combination Antiretroviral Treatment on Liver Metabolic Health in HIV-Infected Persons

**DOI:** 10.3390/v15122432

**Published:** 2023-12-15

**Authors:** Michał Biały, Marcin Czarnecki, Małgorzata Inglot

**Affiliations:** Department of Infectious Diseases, Liver Diseases and Acquired Immune Deficiencies, Wrocław Medical University, 51-149 Wrocław, Poland; marcin.czarnecki@umw.edu.pl (M.C.); malgorzata.inglot@umw.edu.pl (M.I.)

**Keywords:** HIV, cART, ART, antiretroviral therapy, NAFLD, MAFLD, MASLD, NASH, MASH, steatohepatitis, weight gain, insulin resistance, review

## Abstract

In the last three decades, there has been a considerable improvement in human immunodeficiency virus (HIV) therapy. Acquired immunodeficiency syndrome (AIDS) is no longer a common cause of death for people living with HIV (PLWH) in developed countries, and co-infections with hepatitis viruses can be effectively managed. However, metabolic syndrome and metabolic dysfunction-associated steatotic liver disease (MASLD) are emerging threats these days, especially as the HIV-positive population gets older. The factors for MASLD development in PLWH are numerous, including non-specific (common for both HIV-positive and negative) and virus-specific. We focus on what is known for both, and in particular, on the burden of antiretroviral therapy (ART) for metabolic health and liver damage. We review data on contemporary drugs, including different groups and some particular agents in those groups. Among current ART regimens, the switch from tenofovir disoproxil fumarate (TDF) to tenofovir alafenamide fumarate (TAF) and particularly its combination with integrase inhibitors (INSTIs) appear to have the most significant impact on metabolic disturbances by increasing insulin resistance, which over the years promotes the evolution of the cascade leading to metabolic syndrome (MetS), MASLD, and eventually metabolic dysfunction-associated steatohepatitis (MASH).

## 1. Timeline Aspect

### 1.1. The Evolution of Antiretroviral Treatment

In the last three decades, there has been a considerable improvement in human immunodeficiency virus (HIV) therapy—dozens of drugs have been developed, with newer generations consequently gaining on their predecessors [1,2].

The first drugs used in monotherapy lacked efficacy in suppressing viral replication. Subsequently, with the development of successive drug classes, the highly active antiretroviral therapy (HAART) era began, which led to viral suppression. However, those regimens had numerous drawbacks, such as high pill burden, multiple drug interactions, risk of developing drug resistance and treatment-limiting toxicities, including long-term irreversible ones like lipoatrophy (particularly with thymidine analogue NRTIs), bone marrow suppression, hepatotoxicity, and peripheral neuropathy [1,2].

In the new millennium, new generations of drugs have been introduced due to the advancement of research. They no longer require strict food and storage rules, showing a lower pill burden and more favourable toxicities [1]. This has led to well-tolerated single-tablet regimens (STRs), drugs with a high barrier to resistance for people living with HIV (PLWH), and even pre-exposure prophylaxis (PrEP) for those at high risk of infection [1].

There are now approx. 40 anti-HIV drugs of six major, widely available classes. Of the four enzymatic activities found in HIV-1 proteins (protease, reverse transcriptase polymerase, reverse transcriptase, ribonuclease H, and integrase), only ribonuclease H has no approved therapeutical agents targeting it (although effective inhibitors have been found, they show a high level of toxicity and lack selective inhibition) [3]. Also, new drug classes, such as capsid and nucleoside reverse transcriptase translocation inhibitors, have been recently introduced or are in advanced investigation phases [4,5].

As a result of antiretroviral therapy (ART), the mortality rate among PLWH due to acquired immunodeficiency syndrome (AIDS) has decreased over 90 times. Moreover, with better control over the virus, life expectancy among patients rises. In developed countries, even more than 40% of PLWH is over 50, and this value is estimated to reach 73% in 2030. As the drugs are efficient in viral suppression, the issue of improving not only life expectancy but also life quality is raised [6]. Therefore, to the well-known 90–90–90 strategy, another ‘90’ is now added: good health-related quality of life [7].

As attention shifts to the quality of health and life, new antiretroviral (ARV) drugs have to be considered in that aspect, and due to that, they are not completely free of concerns.

### 1.2. Liver Diseases in PLWH over Decades

Liver diseases and liver steatosis in PLWH have been present since the dawn of the HIV pandemic. However, over the last decades, there has been a shift in their origin [8].

The co-infection with primary hepatotoxic viruses (hepatitis B virus (HBV) and hepatitis C virus (HCV)), which have the same transmission route as HIV, was the main reason for hepatic injury. However, at present, those infections can be easily eradicated (HCV) or prevented and controlled (HBV) in developed countries [9,10]. Therefore, when compared, it is evident that the prevalence of HBV/HCV co-infection among PLWH is now significantly lower than in past decades [11], which is also correlated with fewer cases of HIV infections related to injection drug use (IDU) [12].

Abuse of intravenous drugs, apart from potential infection with hepatotropic viruses, can also be toxic to the liver. Hepatopathy associated with intravenous (IV) drugs alone (including opioids) is rare. Nonetheless, drug abuse frequently co-occurs with alcohol abuse, and this combination has a much higher hepatotoxic potential [13].

Although the hepatitis type B and C viruses are no longer a significant threat, liver steatosis in HIV mono-infection remains one. In the pre-antiretroviral therapy era, it was related to malnutrition and opportunist infections. Later, it was linked with hepatotoxicity caused by first-generation NRTIs (such as didanosine (ddI) and stavudine (d4T)). Then, it was associated with the development of metabolic syndrome (MetS) in PLWH, which subsequently leads to metabolic dysfunction-associated steatotic liver disease (MASLD) [14]. It is a serious emerging threat to the health and life of HIV-positive individuals, as shown in statistics. Though the mortality rate among PLWH has generally decreased and AIDS development is no longer a primary concern for the ARV-treated, the death rate due to liver disease complications has increased 8–10 fold compared to the pre-ART era [2] and is among the leading causes of death in PLWH, along cardiovascular diseases and non-AIDS-related neoplasms [15,16].

## 2. Metabolic Dysfunction-Associated Steatotic Liver Disease (MASLD)

### 2.1. Introduction

Formerly known as NAFLD (non-alcoholic fatty liver disease), the disease has been recently renamed and is presently known as MASLD [17]. The new name and definition of the disease encompass the inclusion criteria rather than (as previously) the exclusion of other liver diseases, which is consistent with the current understanding of this disease [17].

It is an umbrella term for a cluster of conditions in a continuum ranging from simple steatosis, inflammatory process (steatohepatitis) to liver fibrosis, cirrhosis, and its entire burden (i.e., liver insufficiency, hepatocellular cancer (HCC), etc.) [18].

MASLD presents with an accumulation of excess lipids in the liver, connected with insulin resistance (IR). It is defined as steatosis of >5% of liver parenchyma. Pathogenetically, it is divided into simple steatosis (formerly called non-alcoholic fatty liver (NAFL)) and metabolic dysfunction-associated steatohepatitis (MASH) [19]. Although the natural history of MASLD remains mild and involves simple steatosis for many years, MASH may develop over time in about 20–40% of cases. In addition, about 23–35% of inflamed livers will develop liver fibrosis. Among those, 9–20% will subsequently progress to liver cirrhosis and its complications [20]. In the USA, the incidence rates of HCC increased from 4.4 (in 2000) to 6.7 (in 2012) per 100,000. Also, HCC has been the fastest-rising cause of cancer-related deaths in the USA [21].

Until recently, NAFLD diagnosis required excluding other factors that might result in liver steatosis (alcohol abuse, viral hepatitis, hepatotoxic drugs, and other liver diseases) [19]. However, overlapping of those circumstances may be present [19]. Therefore, in the new consensus, simplified criteria for MASLD diagnosis were stated, including steatosis found by imaging or biopsy and the presence of at least one of the five cardiometabolic risk factors (CMRFs) [17]. In case of doubt, the final diagnosis depends on the liver biopsy histopathological result [19].

The incidence of MASLD among lean people (hitherto ‘lean NAFLD’) is defined as the presence of hepatosteatosis in a person with a normal body mass index (BMI), excluding the factors mentioned above, for example, alcohol abuse. This condition occurs in approx. 20% of MASLD cases in Europeans and approx. 5–45% MASLD cases in Asians. In the general population (i.e., non-infected with HIV), dissimilarity in pathogenesis is underlined (including the role of microbiota and individual metabolic factors) [18,22]. This term is specifically relevant for MASLD in PLWH as the percentage of affected is higher than in the general population (35.4% and 24.2%, respectively) [2].

### 2.2. MASLD Pathogenesis in the General Population

The progression of MASLD from simple steatosis, through MASH, to HCC was originally described as a two-hit model involving the initial development of steatosis (conditioning susceptibility to further damage) and a second factor triggering the exacerbation of lipid peroxidation and inflammation. However, based on the evidence, the model has been rearranged to a multi-hit or multi-parallel hit model, including multiple pathways promoting progressive fibrosis and oncogenesis. This model includes multiple cellular, genetic, immunological, metabolic, and endocrine factors [23].

The main factor is overnutrition, which leads to obesity. This is followed by the development of MetS and comorbidities, including type 2 diabetes mellitus (T2DM), hypertension (HT), hyperlipidaemia (HL) or dyslipidaemia (DL), chronic kidney disease (CKD), cardiovascular disease (CVD), obstructive sleep apnoea (OSA), osteoarthritis, malignancies (e.g., of the breast, colon, and prostate), and MAFLD [24]—which is the main subject of this paper. Still, it is difficult to speak about it in isolation from the rest of the components of metabolic syndrome. Also, irrespective of MASLD, diabetes and obesity increase HCC risk [21].

A link between MASLD and T2DM appears evident. These two pathologic conditions frequently coexist, and there is a bidirectional correlation, as obesity and insulin resistance are key pathogenic factors for both. Previous studies prove that T2DM is an established risk factor for the progression from simple steatosis to MASH and cirrhosis, but the presence of MASLD might also precede and promote T2DM development [25].

Adipose tissue is not only an energy supply but also an important hormone-secreting tissue (producing adipokines and lipokines). Its accumulation leads to disturbances of hormonal activity—hypertrophic adipocytes present in obese individuals show increased production of pro-inflammatory cytokines and reduced production of anti-inflammatory cytokines, such as adiponectin, affecting the insulin sensitivity of tissues. This leads to the growth of insulin resistance [18,26]. In turn, this results in impaired lipolysis and increased lipogenesis, causing a metabolic overload of the liver. The accumulation of lipids in hepatocytes restrains their oxidative capacity, subsequently promoting lipid oxidation. The lipotoxic lipids induce cellular stress, followed by increased reactive oxygen species (ROS), thus promoting interleukin 6 (IL-6) and cytokeratin 18 (CK-18) production and stimulating cell apoptosis and fibrogenesis [2,18].

Another factor to be considered is gut microbiota alterations. Dysbiosis is associated with the development of MASLD. However, it is hard to precisely describe the complex interplay between gut microbiota, its metabolites, and MASLD progression [27].

In addition, a certain genetic component has been attributed to MASLD; several genetic variants have been described, though the best defined and associated with the development of the disease are single nucleotide polymorphisms (SNPs) in the patatin-like phospholipase domain-containing 3 (PNPLA3) gene. However, this and other variants account for a small number of cases and produce a synergistic effect with environmental factors [18,22].

### 2.3. PLWH-Specific Factors for MAFLD Development

As mentioned, liver steatosis is widespread in the general population and PLWH. In the latter, it has a higher prevalence, not infrequently more severe course, and is associated with specific aetiopathogenetic factors [2].

The general population’s MASLD prevalence ranges from 13.5% in Africa to 31.8% in the Middle East [18]. It occurs in 47.3–63.7% of patients with T2DM and up to 80% of individuals with obesity. In comparison, MAFLD occurs in about 35–48% of PLWH (though some studies report a range of 13–73%) [28,29].

The pathogenesis of MASLD in PLWH is a consequence of classic pathogenetic factors (mindful of their different severity in this group) on the one hand, but also a specific set of factors associated with HIV on the other hand [2]. Mentioned here are MetS (HT, DL, waist circumference (WC), and IR), HIV-associated lipodystrophy, hyperuricaemia, ART, HIV (the virus itself), and gut microbiota [30].

A higher steatosis incidence was observed in histopathological studies in PLWH not treated with ART [20]. Though HIV is not considered a classic hepatotropic virus, its envelope interacts with hepatocytes, promoting ROS production and oxidative stress. Also, macrophage activation occurs. This enhances inflammation and fibrogenesis following tissue regeneration [7]. Long-time infection and even minimal HIV replication cause low-grade chronic inflammation and constant minimal immune stimulation, promoting faster ageing of the body and thus metabolic burdens associated with ageing [31]. The term ’inflamm-ageing’ has been coined to describe it [30].

HIV infection also affects the architecture of the intestinal wall and the gut microbiota composition. This promotes the translocation of intestinal bacteria and the constant stimulation of the host’s immune system. In addition, PLWH suffer from the impoverishment of microbiota diversity—which appears to be quite similar to that found in obese individuals (i.e., a greater share of Enterobacteriaceae and fewer Bacteroidetes and Firmicutes). ART is unlikely to restore intestinal diversity, as found in uninfected individuals. Translocation of bacteria, especially lipopolysaccharides, increases inflammatory processes in adipose tissue and subsequently promotes the production of IL-6, tumour necrosis factor (TNF), and other pro-inflammatory cytokines [26].

The advancement of immunodeficiency, referred by nadir CD4+ T-cell count, seems to be another risk factor for MASLD [29].

Last but definitely not least, aspects of the MASLD pathogenesis in PLWH are the adverse effects of ART, which may contribute to the development of liver steatosis [29]. In the old generations of drugs (especially thymidine-derived NRTIs such as zidovudine (AZT), ddI, and d4T), lipodystrophy was often accompanied by dyslipidaemia and liver steatosis or steatohepatitis [32]. Currently, the mechanism by which modern drugs promote the development of metabolic disorders is more complex and is the subject of ongoing research [33].

## 3. Impact of Antiretroviral Agents

### 3.1. General Approach

Several mechanisms are to be listed in the matter of ART-induced liver damage, including hypersensitivity reactions (idiosyncratic hepatotoxicity) through direct mitochondrial inhibition, direct cell stress, or immune reconstitution, particularly in the presence of viral hepatitis co-infection, lipid/carbohydrate metabolism disturbances, and steatosis as a separate mechanism [3].

In the 1990s, after introducing several antiretroviral drugs, the model therapy consisted of three agents—two nucleoside reverse transcriptase inhibitors (NRTIs) and a protease inhibitor (PI) (such as ddI, d4T and indinavir (IDV), nelfinavir (NFV), and ritonavir (RTV), respectively). Unfortunately, the undesirable outcome was lipodystrophy in up to 50% of patients. Nevertheless, new generations of antiretroviral agents, despite having a much safer metabolic profile and not causing severe lipodystrophy, appear to promote metabolic disturbances, resulting in increased BMI and the development of the components of MetS [31,34].

Assessing the impact of particular substances is difficult due to their use in combined therapy. Therefore, the effects (including adverse) of treatment that can be observed are the cumulative work of several agents applied simultaneously.

Below, the different groups of drugs and their impact on liver health are described.

A summary of the proposed metabolic effects of ARV drugs in the main ART groups is shown in Table 1.

### 3.2. NRTIs

The first generation of NRTIs (zalcitabine (ddC), ddI, d4T, and AZT) is no longer used in developed countries. These drugs present mitochondrial toxicity. They inhibit not only the HIV reverse transcriptase but also mitochondrial gamma polymerase, which affects mitochondrial divisions, providing impaired mitochondrial function and increased lactate production as well as abnormal lipid oxidation in the liver, leading to steatosis and other lipid disorders [14].

The second generation of NRTIs, including tenofovir, has a much safer profile [35].

Interestingly, tenofovir disoproxil fumarate (TDF) is associated with an improved lipid profile. In comparison, tenofovir alafenamide (TAF) seems to have no effect on serum lipids. However, switching from TDF to TAF results in increased serum lipids, and switching back to TDF decreases those parameters, indicating the specificity of the effect of TDF [31]. Therefore, it is suggested that TDF and other non-TAF NRTIs suppress appetite. In the case of TDF, other mechanisms affecting the lipid profile are also suspected, but they are still unknown [36,37].

Another study regarding metabolic and liver functions in 190 patients showed that within six months after switching from TDF to TAF, there was a decrease in the activity of liver enzymes, with a simultaneous increase in glucose (Glc), total cholesterol (TC), high-density lipoprotein (HDL), and low-density lipoprotein (LDL) [38].

In a study group of >4000 people who switched from TDF to TAF regimen, after 18 months, the lipid profile deteriorated, and overweight was observed in the switch group. This was regardless of whether TAF was combined with PIs, non-nucleoside reverse transcriptase inhibitors (NNRTIs), or integrase inhibitors (INSTIs). This study ruled out the recovery effect (which may result in weight gain after starting ART) because all study participants had already been treated. Although weight gain and dyslipidaemia favour IR, the study results demonstrated no clear evidence that DM is more common in TAF (during 18 months of this study). The weight gain effect was also noticed after switching from abacavir (ABC) to TAF [37].

Changing the regimen from ABC to TAF resulted in an increase in body weight, but it was less significant than in the switch from TDF to TAF. There is a proposal that ABC also has a weight-suppressive effect. It remains unclear whether this ABC property is a side effect (average drug tolerance) that affects appetite and caloric intake or whether it impacts mitochondrial and adipocyte function [36]. Because TAF shows better tolerance than ABC, it may not bring this effect in such an interpretation [39].

Another study involving 146 individuals indicated that participants exposed to TAF were twice as likely to have steatosis as those unexposed [40].

In a meta-analysis of eight randomized clinical trials (RCTs), TAF/emtricitabine (FTC) was associated with the highest weight gain, ABC/lamivudine (3TC) and TDF/FTC with a slightly lower weight gain, and AZT/3TC with weight stability [39].

In an analysis comparing ATV and NRTIs (TAF, ABC, and TDF), the first showed lower weight gain than the others. However, among these, ATV is the most toxic for the gastrointestinal tract (GIT), which may result in lower caloric intake [39].

PrEP studies provide exciting results showing that individuals unburdened by HIV infection are exposed to dual NRTI therapy. For example, a study evaluating the effect of PrEP with TDF/FTC vs. placebo on weight gain found a lower weight gain in the TDF group, suggesting an effect of inhibiting mass growth by TDF; in turn, when comparing PrEP TAF/FTC and TDF/FTC regimens, after 48 weeks, there was an increase in weight by 1.1 kg in the TAF group and no increase in the TDF group [39].

### 3.3. NNRTIs

First-generation NNRTIs have been associated with a number of mechanisms that can lead to liver damage, including mitochondrial toxicity and hypersensitivity. However, second-generation NNRTIs seem to be liver-safe [3].

Nevirapine (NVP) is a remarkably hepatotoxic drug. A significant number of patients exposed to NVP develop short-term hypersensitivity reactions, which can manifest as hepatotoxicity, for example. Also, late-onset hepatotoxicity is common. The mechanism is not fully understood; however, damage-associated molecular patterns (DAMPs) and subsequent immune activation triggering with coexisting mitochondrial dysfunction are proposed to be the reason for hepatocyte death [3].

Efavirenz (EFV) has been associated with an increased risk of progressive liver steatosis in HIV/HCV co-infections. In mice, it induces liver steatosis in a hepatic pregnane X receptor (PXR)-dependent manner. PXR is known for regulating the expression of genes mediating cholesterol biosynthesis [14]. EFV also shows a relatively high concentration in adipocytes [41]. Higher EFV concentrations have been shown to cause cell stress in the liver by acutely inhibiting mitochondrial function and leading to the accumulation of fatty acids (FAs) and TG in the cytoplasm [28]. The mitochondrial toxicity of EFV has been proposed as a reason for the BMI increase [42]. On the other hand, it has been connected with appetite suppression [36].

Rilpivirine (RPV) seems to have a good safety profile compared to other NNRTIs, like EFV, regarding lipid abnormalities and blood TC and TG. It has been observed that RPV also prevents excessive lipid droplet accumulation. Also, some cases of improved lipid profiles after treatment with RPV have been reported [43]. Perhaps the hepatoprotective role of RPV is associated with an alteration of autophagy, which mitigates hepatic stellate cell activation [44].

Doravirine (DOR) is a novel third-generation NNRTI. In a study comparing DOR with darunavir boosted with ritonavir (DRV/r), the first had better LDL and non-HDL cholesterol profiles than the latter. DOR shows at least a slight improvement in the lipid profile, where DRV/r deteriorates it [45]. Another analysis demonstrates a comparable mean weight gain in the DOR, DRV/r, and EFV groups (1.7, 1.4, and 0.6 kg after 48 weeks, respectively). In addition, the proportion of individuals whose BMI increased to overweight or obese or those who noted at least a 10% weight gain did not differ significantly between the groups [46]. Also, short-term liver toxicity for DOR seems uncommon [45].

### 3.4. PIs

Exposure to older protease inhibitors (such as IDV and lopinavir (LPV)) may accelerate the development of metabolic syndrome components through multiple pathways, including exacerbated mitochondrial dysfunction, increased ROS and pro-inflammatory cytokine production, and impaired function of glucose transporters (GLUTs), resulting in the development of IR [47], which is the main pathogenetic factor leading to MASLD/MASH [28].

In a cross-sectional study involving 136 HIV-positive patients, the use of LPV was significantly associated with an increased controlled attenuation parameter (CAP) in transient elastography, while DRV showed a negative correlation with CAP [48].

In an analysis of several RCTs (>1000 individuals), insulin sensitivity did not show significant changes with ATV, whether this study involved healthy individuals for a short term or HIV-infected individuals for the long term in combination with other drugs, like tenofovir and FTC, parallelly. LPV had increased IR, and DRV did not demonstrate any significant changes in insulin resistance in HIV-infected individuals in long-term use in combination with another drug [49].

In the case of the deterioration of the lipid profile, the mechanism by which PIs may cause it is vague. It is suggested that an increase in the activity of apolipoprotein B, which transports circulating LDL, very-low-density lipoprotein (VLDL), and TG, may be involved. A study in mice showed that RTV inhibits the clearance of TG from circulation [14].

In terms of weight loss or gain, the issue of gastrointestinal disorders (subsequently resulting in reduced calorie intake) due to drugs’ adverse effects is often raised. A large (>16,000 individuals) cohort study compared currently used INSTIs and DRV in terms of gastrointestinal disorders and showed no difference in ART-naïve patients, though DRV in the switch group appeared to trigger adverse effects in the gastrointestinal tract more often than some INSTIs [15].

### 3.5. INSTIs

INSTIs are the newest widespread drug class. They show favourable pharmacokinetics and efficacy and are generally well tolerated by PLWH. They are currently preferred components for individuals initiating therapy. However, numerous data indicate they are associated with more weight gain than NNRTIs and PIs [50]. The mechanism of INSTI-related weight gain remains unclear. Several explanations are suggested [41]. Also, there seem to be noteworthy differences among various INSTIs and backbone drugs—NRTIs (particularly TAF) as the most considerable [41].

INSTIs potentially affect adipocyte differentiation, thermogenesis, and oestrogen-dependent metabolic pathways; it is proposed that they may impact the secretion of adiponectin or other hormones regulating glucose and lipid metabolism. In addition, they seem to inhibit the stimulation of melanocortin, which may increase appetite. They also have a potential for magnesium chelating, which promotes insulin resistance [41].

A decrease in resting energy expenditure (REE) is also proposed due to the rapid suppression of viral load, which may explain the greater increase in BMI in treatment-naïve individuals. However, weight gain is also noticeable in patients with suppressed viral load prior to the onset of INSTI treatment [41,51].

Furthermore, it is hypothesized that weight gain and metabolic disturbances are secondary to microbiota disturbances; however, no direct mechanism has been revealed. INSTIs have been associated with microbiota modification and subsequent changes in the concentration of fatty-acid-binding proteins (FABPs). This is known to be an independent predictor of weight and visceral adipose tissue gain in ART-treated individuals [41,51].

It has been found that dolutegravir (DTG) has a potential for inhibiting melanotropin (MSH) binding to melanocortin 4 receptor (MC4R), which affects leptin signalling responsible for food intake (its deficiency is linked with monogenic obesity) [14,39]. Nonetheless, another study indicates the lack of dependency between the grade of this effect and drugs’ potential for body mass increase (e.g., elvitegravir (EVG) is a 25-times more potent MC4R antagonist and has a lesser impact on weight gain). Furthermore, this phenomenon is observed in vitro, with drug concentrations multiple times exceeding therapeutic ones. Most probably, INSTIs affect other metabolic pathways, albeit this has not yet been confirmed [52].

In the case of EVG, the mechanism of causing oxidative stress connected with fibroblast growth factor 21 (FGF21) and betaKlotho is described. The former is a marker of metabolic disturbances in individuals with obesity or DM, and the latter impairs glucose uptake. Other INSTIs are neutral for those substances [53].

Ultimately, the weight gain connected with the group may be due to the effects of INSTIs and/or the fact that one is treated with an INSTI regimen instead of an appetite suppressant (e.g., containing EFV) [36].

A cohort study with >16,000 individuals has shown that a moderate-to-high liver parameter increase in INSTI-treated is uncommon. There are also no significant differences among DTG, EVG, and raltegravir (RAL) in that area. The hepatotoxicity of each of those drugs is low [15].

In another study on the hepatosteatosis effect of INSTIs, exposure to EVG and RAL was associated with higher odds of moderate-to-severe hepatic steatosis; no association was found with exposure to DTG. Furthermore, the relationship with EVG was consistent regardless of the combination with TDF or TAF [54]. Also, other authors report that DTG has no impact on liver steatosis [40].

In the case of weight gain, a meta-analysis of eight studies from 2022 proves that relevant weight gain is to be observed mainly in therapy with DTG (79.2%) and BIC (77.9%). Its probability is less significant when RAL (33.2%) or EVG (9.7%) are used [51].

Also, in a prospective cohort study lasting five years, INSTIs (without specifying which ones) were linked with hepatic steatosis in the context of increased body weight gain [29].

On the other hand, a recent Spanish study (including almost 5,000 patients) has proved that being exposed to INSTIs was associated with a lower risk for hepatic steatosis (measured with the Hepatic Steatosis Index (HSI)) [55].

Cabotagravir (CAB), a novel INSTI, dosed as a long-acting injection, has shown a favourable metabolic profile in PrEP vs. placebo study—with no interference with body weight or glucose and lipid profile [56]. Furthermore, another study comparing CAB/RPV vs. DTG/RPV has indicated no hepatic steatosis onset in the CAB arm [57].

### 3.6. CCR5 Receptor Antagonists (aCCR5)

Data on MVC concerning liver and metabolic parameters usually come from in vitro or murine studies. These studies prove it has the potential to regulate genes responsible for cytokine release and affect microbiota, which may result in BMI reduction and insulin sensitivity improvement [14,58,59]. However, drug-induced liver injury (DILI) has been reported in humans, and another agent from this group has been withdrawn from further development due to prominent hepatotoxicity [3].

### 3.7. New Drug Classes

Lenacapavir (FDA-approved in December 2022) is a novel, first-in-class, multi-stage, selective inhibitor of the HIV capsid protein. In the short term, carbohydrate metabolism disturbances occur infrequently [60]. In a recent update from week 80 of a phase 2 trial with co-administration of other ARV agents, a noticeable weight gain has been shown (+4.7 kg) only in arms with added TAF; however, in arms with no TAF, the gain has been milder (+2.6 kg) [61].

Islatravir (in development) is a long-acting, first-in-class nucleoside reverse transcriptase translocation inhibitor (NRTTI) with versatile dosing routes and intervals; so far, it has shown good efficacy in the short-term, showing a low risk of triggering severe hypertriglyceridaemia and hypertransaminasaemia [62,63,64].

### 3.8. Intergroup Comparison

In a study comparing regimens including FTC with a combination of TAF/DTG, TDF/DTG, and TDF/EFV in >1000 treatment-naïve patients, DTG, especially when combined with TAF, affects body mass increase greater than EFV. Weight and visceral adipose tissue gain in the TAF arm have been twice as high as in the EFV arm [42].

A switch from boosted PIs (DRV, ATV, and LPV) to INSTIs (RAL and DTG) regimens has shown an improvement in insulin sensitivity, reduction of insulin (and homeostatic model assessment—insulin resistance (HOMA-IR)) and leptin levels. No difference between these two INSTIs has been revealed. However, this result is inconsistent with other studies that showed no significant difference in HOMA-IR among regimens with TDF/FTC with ATV/r, DRV/r, or RAL [6].

Another study with a switch from EFV to RAL (with an unchanged dual NRTI backbone) reports hepatosteatosis mitigation in the latter group [28]. Also, a study with a switch from DRV-, ATV-, and LPV-based regimens to RAL has demonstrated improvement in steatosis [6].

However, a meta-analysis of >5000 PLWH indicates that INSTIs have a greater weight gain potential than NNRTIs or PIs [39]. In the comparison of ATV and NRTIs (TAF, ABC, and TDF), the former has shown a lower gain than the latter. However, among these, ATV is the most toxic for the gastrointestinal tract (GIT), which may reduce caloric intake [39].

## 4. Conclusions

Undoubtedly, hepatic steatosis has a multifactorial background. A large part of the circumstances underlying this condition has been revealed; however, the exact impact of each is yet to be estimated.

In the general population, the presence of steatosis reflects metabolic disturbances (both in lipids and carbohydrates). The presence of T2DM or prediabetic state often coexists or precedes the appearance or progression of hepatosteatosis. In PLWH, the aforementioned components common for both HIV+ and HIV− people (including major ones such as metabolic syndrome, overnutrition, and probably minor ones such as gut dysbiosis and genetic factors) have to be considered, as well as the factors specific to PLWH (drug adverse effects (AEs), chronic immune activation, and additional gut immune tissue and microbiota alterations). From the latter, the antiretroviral drugs seem relevant in pathogenesis, though estimating their precise influence remains unlikely.

New therapeutic regimens show distinctly more favourable safety profiles than their predecessors. They are less hepatotoxic, so direct liver damage is sporadic, albeit they (or at least some of them) have the potential to affect metabolic disorders and their development. Apart from the mechanism, it is relevant to distinguish the ‘return to health’ effect (of treatment) from unwelcomed weight gain.

Among current ART regimens, INSTIs used with TAF appear to have the most significant impact on metabolic disturbances by increasing insulin resistance, which over the years, promotes the evolution of metabolic syndrome components, eventually triggering inflammatory processes, accelerating ageing, and leading to steatohepatitis. For INSTIs alone, the current literature presents discordant data.

Studies are emerging, focusing not only on lipid abnormalities but also on carbohydrate metabolism (which should be considered important in the development of metabolic disturbances in PLWH or ART patients). This will, hopefully, bring improvement in the care of HIV-positive individuals. In addition, the two-drug regimen trials show good results, predicting a new paradigm promising fewer drug adverse effects and interactions [65].

Most of the reviewed studies lasted 48 or 96 weeks. Also, the trials (including safety outcomes) for drugs usually last 48 or 96 weeks [66,67,68], as recommended by the Food and Drug Administration (FDA) for developing antiretroviral (ARV) drugs [69]. Because direct hepatotoxicity of concurrent ARV drugs is low or unnoticeable, it often takes years to develop metabolic complications. Therefore, the most valuable studies are those with long follow-up periods, which comprise a small number of ARV drug studies.

Last but not least, the importance of lifestyle needs to be underlined [19,70]. Many factors must be included in MASLD pathogenesis, but the modifiable factors have an important role. Lifestyle modification, including caloric restriction (500–1000 kcal/day, targeting gradual weight loss of 7–10% in overweight/obese persons), restraint in simple carbohydrate intake, Mediterranean diet, and moderate aerobic exercise (150–200 min a week during 3–5 sessions) [71,72] should be an essential part of a holistic approach to patient’s therapy—unfortunately, the physician’s involvement alone in this regard is often insufficient. Patients should be under the care of an entire team of multidisciplinary specialists, including dietetic, psychological, and physical effort aspects.

## Figures and Tables

**Table 1 viruses-15-02432-t001:** A summary of proposed ARV agents’ metabolic effects.

Group of ARV Agents	Agents	Metabolic Effects	Additional Information
NRTIs	First generation	ddC, ddI, d4T, and AZT	High mitochondrial toxicity due to the inhibition of mitochondrial gamma polymerase	No longer in common use
Second generation	TDF	Appetite suppression, improvement of lipid profile	
TAF	No effect or increase of serum lipids; weight gain—peculiarly when switched from another NRTI agent, especially from TDF; effects particularly possible in combination with INSTIs	
ABC	Probable weight suppression effect	
NNRTIs	First generation	NVP	Hepatic and mitochondrial toxicity; hypersensitivity reactions; suggested DAMP pathway in hepatocyte death	No longer in common use
EFV	Suggested PXR-dependent manner of causing steatosis; accumulation of FAs and TGs in cytoplasm; has been connected with appetite suppression	No longer in common use
Second generation	RPV	Relatively good metabolic safety, potential hepatoprotective effect (due to mitigation of hepatic stellate cell activation) and even possible improvement in the lipid profile (prevents excessive lipid droplet accumulation)	
Third generation	DOR	Provides slight improvement in the lipid profile, comparable weight gain with DRV	
PIs	First generation and older second generation	INV, IDV, RTV, LPV, and ATV	Mitochondrial dysfunction, increased ROS production; impairment of GLUTs—resulting in IR development	No longer in common use, except for RTV as a booster
Newer second generation	DRV	No or non-significant impact on IR;possible GIT toxicity causing weight loss	
INSTIs	DTG, EVG, and RAL	Potential changes in adipocyte metabolism and adipokine secretion; increase of IR; lack of GIT AEs—good appetite and greater calorie intake	DTG seems to show the greatest weight gain;higher weight gain, in particular, when used in combination with TAF
CAB	Shows favourable metabolic profile (no interference with weight gain or glucose and lipid profile); possibly no effect on liver steatosis induction

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
