# Peer review of "Impact of Combination Antiretroviral Treatment on Liver Metabolic Health in HIV-Infected Persons"

_viruses, 2023, doi:10.3390/v15122432_

Round 1

Reviewer 1 Report (Previous Reviewer 3)

Comments and Suggestions for Authors

The manuscript has been revised sufficiently.

Reviewer 2 Report (Previous Reviewer 1)

Comments and Suggestions for Authors

The authors have adequately responded to my concerns and those of the other reviewers.

Comments on the Quality of English Language

/

This manuscript is a resubmission of an earlier submission. The following is a list of the peer review reports and author responses from that submission.

Round 1

Reviewer 1 Report

Comments and Suggestions for Authors

Due to huge improvements in HIV therapy over the last three decades people with HIV typically no longer die of AIDS but liver disease is an emerging concern. The title, “Impact of cART on liver metabolic health in HIV infected persons” led me to believe the focus would be on antiviral therapy.

I found the authors used too much space on things that are not particularly relevant to the title, such as the third and fourth paragraphs in section 1.2 about the changing epidemiology of HBV and HCV, and did not give enough attention to the main topic. If the intent was a comprehensive review of liver related epidemiology then the title should be changed to reflect that.

 I think there is an error at line 220 “In the 1990s, after introducing several antiretroviral drugs, the model therapy consisted of three agents – two non-nucleoside reverse transcriptase inhibitors (NNRTIs) and a protease inhibitor (PI)” I believe this should be nucleoside reverse transcriptase inhibitors (NRTI.)

Comments on the Quality of English Language

Writing does not follow standard English composition with a topic sentence and supporting evidence.  For example, Section 3.2 has many one and two sentence paragraphs. At line 246 the claim is made. “Therefore, it is suggested that TDF and other non-TAF NRTIs suppress appetite.”  The supporting evidence for this claim is scattered about the remainder of the section. 

Several vague statements are made without complete information.  Two examples are:

Line 94 “the death rate due to liver disease complications has increased 8–10 fold [2]”  Compared to what?

Line 242 “decreased lipid parameters” Line 244 “increased lipid parameters”  What does this mean?

 Abbreviations are used without first defining them.  Two examples are:

INSTI, first used at line 255, is never defined in the body text, only in the abstract.

aCCR5 used at line 398 is never defined.

Author Response

The mentioned paragraphs from section 1.2 were meant to underline the evolution of liver injury etiology, up to current point. However, we consider the comment as accurate and those paragraphs have been reduced.
There was an error at line 220. Indeed, there should be “NRTIs”, not “NNRTIs”. It has been corrected.
Abbrevations (INSTIs, aCCR5) used without first defining them in the body have been supported by the full names.
Line 94 (currently 83) – the sentence has been extended (“compared to pre-ART era”).
“Increased/decreased lipid parameters” have been reworded to “increased serum lipids” and “improved lipid profile”.
Also, the text has gone through an another round of a language review by a native speaker.

The manuscript has been resubmitted with the changes mentioned above.

Reviewer 2 Report

Comments and Suggestions for Authors

The present review article provides an overview of how combination antiretroviral therapy (cART) affects the metabolic health of individuals living with HIV. The manuscript is skillfully composed and effectively encapsulates key facets of liver-related issues in HIV. While I have no major reservations about the manuscript, I offer one suggestion: it would enhance the reader's comprehension if the author included relevant visuals to aid in conveying information more effectively.

Comments on the Quality of English Language

Minor editing require..

Author Response

The point of our review was to show effects of contemporary used antiretroviral drugs on liver metabolic health. Therefore we created a Table 1 with the effects are collected. We believe it is simple yet clear summary of the key point of our paper.
The text has gone through an another round of a language review by a native speaker.

The manuscript has been resubmitted with the changes mentioned above.

Reviewer 3 Report

Comments and Suggestions for Authors

General:

This article was intended to review the latest clinical literature related to the factors leading to MASLD/NAFLD development in HIV-infected patients. Special emphases were placed on the impact of different ART regimens currently used.  In view of several existing recent reviews related to the subject, the merits of the new information appear somewhat modest but maybe sufficient; some wordings/statements are unusual or unclear and certain conclusions maybe controversial.

Specifics:  

1.       It was summarized in the abstract that “Integrase inhibitors (INSTIs) appear to have the greatest impact on metabolic disturbances” However, in the section describing the effects from the individual INSTIs, both positive and negative effects were presented, and the findings from the literature for both sides seem to be equally credible. Therefore, it could be premature or misleading to conclude that INSTIs would have the greatest impact on metabolic disturbances (except clearcut individual agents).  It is also recommended the latest publication by Navarro el al, Prevalence of non-alcoholic fatty liver disease in a multicenter cohort of people living with HIV in Spain.  Eur J Intern Med. 2023; 110:54-61, should be included which indicates that HIV patients exposed to INSTI showed a lower risk of NAFLD.

2.       The negative impact from TAF, especially after switching from TDF to TAF regimen is unique and important. And the impact was not limited to combinations with INSTIs, but with all other ART groups.  Such a factual findings and implication deserves to be emphasized in the abstract. Table 1 should be revised to reflect the balanced emphasis. 

3.       Several wordings through the manuscript are recommended to be changed to more appropriate/scientific words/phrasings; e.g., ln 160, “undermines their oxidative capacity”, exact meaning is unclear; ln 162: IL6, CK 18 are NOT ROS species; ln 175 “bigger” and “harder” should be replaced; ln 447, “yet unlikely”?  456, “biggest synergy”?  ln 76, “make a big part”? 

Comments on the Quality of English Language

Revision for unusual words/phrasings are recommended.

Author Response

The data from the publication by Navarro et al. has been incorporated into our text.
The impact of switching from TDF to TAF in matter of metabolic changes has been emphasized both in the abstract and in the table 1.
The incorrect classification of CK-18 and IL-6 as ROS has been rectified.
The text has gone through another round of language review by a native speaker.

The manuscript has been resubmitted with the changes mentioned above.
